# *Actinobacillus succinogenes* in Bioelectrochemical Systems: Influence of Electric Potentials and Carbon Fabric Electrodes on Fermentation Performance

**DOI:** 10.3390/microorganisms13081720

**Published:** 2025-07-23

**Authors:** Julian Tix, Jan-Niklas Hengsbach, Joshua Bode, Fernando Pedraza, Julia Willer, Sei Jin Park, Kenneth F. Reardon, Roland Ulber, Nils Tippkötter

**Affiliations:** 1Bioprocess Engineering and Downstream Processing, University of Applied Sciences Aachen, 52428 Jülich, Germany; tix@fh-aachen.de (J.T.); bode@fh-aachen.de (J.B.);; 2Mechanical and Process Engineering, RPTU Kaiserslautern-Landau, 67663 Kaiserslautern, Germany; janniklas.hengsbach@rptu.de (J.-N.H.); roland.ulber@mv.rptu.de (R.U.); 3Kynol Europa GmbH, 22453 Hamburg, Germany; j.willer@kynol.de; 4Chemical and Biological Engineering, Colorado State University, Fort Collins, CO 80523, USA; s.park@colostate.edu (S.J.P.); kenneth.reardon@colostate.edu (K.F.R.)

**Keywords:** *Actinobacillus succinogenes*, succinate, lactate, bioelectrochemical systems, carbon fabric electrodes

## Abstract

The fermentation of *Actinobacillus succinogenes* in bioelectrochemical systems offers a promising approach to enhance biotechnological succinate production by shifting the redox balance towards succinate and simultaneously enabling CO_2_ utilization. Key process parameters include the applied electric potential, electrode material, and reactor design. This study investigates the influence of various carbon fabric electrodes and applied potentials on product distribution during fermentation of *A. succinogenes*. Building on prior findings that potentials between −600 mV and –800 mV increase succinate production, recent data reveal that more negative potentials, beyond the water electrolysis threshold, trigger electrochemical side reactions, altering product yields. Specifically, succinate decreased from 19.76 ± 0.41 g∙L^−1^ to 14.1 ± 1.6 g∙L^−1^, while lactate rose from 0.59 ± 0.12 g∙L^−1^ to 3.12 ± 0.21 g∙L^−1^. Contrary to common assumptions, the shift is not primarily driven by oxygen formation. Instead, the results indicate that the intracellular redox potential is affected by both the applied potential and hydrogen evolution, which alters metabolic pathways to maintain redox balance. These findings demonstrate that more negative applied potentials in electro-fermentation processes can impair succinate yields, emphasizing the importance of fine-tuning electrochemical conditions in the system for optimized biotechnological succinate production.

## 1. Introduction

### 1.1. Succinate

Succinate is considered a key platform chemical of the future. In particular, the possibility to produce succinate by fermentation and to fix CO_2_ in the citrate cycle (TCA) is a promising approach [1,2]. According to the U.S. Department of Energy, succinic acid is one of the twelve platform chemicals with the greatest potential for biotechnological production from biomass [3]. Succinate is utilized in a variety of fields. For instance, it is employed in the synthesis of surfactants and detergents, as a food additive (E 363), in the field of medicine, and in numerous other domains. However, succinates are also employed in the production of basic chemicals, such as 1,4-butanediol, 2-pyrrolidinone, γ-butyrolactone, and polymers, such as the biodegradable polymer polybutylene succinate [4].

### 1.2. Actinobacillus succinogenes

*A. succinogenes* is a facultative anaerobic, capnophilic, Gram-negative, biofilm former that is non-pathogenic and exhibits pleomorphic characteristics [5,6,7]. It belongs to the genus *Actinobacillus* and was first isolated from the bovine rumen [7]. Its optimal growth conditions occur at a pH of 7 and a temperature range of 37–41 °C [8]. *A. succinogenes* metabolizes glucose into various organic compounds, including succinate, acetate, formate, ethanol, and lactate [9]. The metabolism of *A. succinogenes* is characterized by an incomplete tricarboxylic acid (TCA) cycle, with three primary pathways contributing to succinate formation: the glyoxylate pathway, as well as the oxidative and reductive branches of the TCA cycle (C4 pathway). In this bacterium, only the reductive branch is active in succinate production [10]. Additionally, outside the TCA cycle, the C3 metabolic pathway facilitates the formation of by-products such as lactate, formate, acetate, and ethanol [11]. Pyruvate serves as a central intermediate, feeding into multiple metabolic pathways. Within the TCA cycle, pyruvate is converted to succinate via phosphoenolpyruvate (PEP) and oxaloacetate (OAA). PEP is metabolized by phosphoenolpyruvate carboxykinase (PEPCK) to form OAA, which is then reduced to malate by malate dehydrogenase. Malate undergoes dehydration to fumarate by means of fumarase (FUM) and is subsequently reduced to succinate by succinate dehydrogenase (SDH). Throughout this process, ATP and NADH function as energy and reducing agents, respectively [9]. Succinate production by *A. succinogenes* occurs only under anaerobic conditions, as fumarate serves as the final electron acceptor exclusively in the absence of oxygen. Under aerobic conditions, oxygen (O_2_) takes over this role, preventing succinate production and enabling the bacterial electron transport chain to function normally [12]. The overall reaction from monosaccharide breakdown via glycolysis to succinic acid results in a net consumption of one mole of NADH. To maintain redox balance, this NADH must be regenerated [13]. The theoretical yield of succinic acid using glucose as substrate and the presence of CO_2_ is 1.71 mol_succinate_·mol_glucose_^−1^, which corresponds to 1.12 g_succinate_·g_glucose_^−1^ [14]. The cofactors NAD^+^ and NADH play essential roles as electron carriers in numerous metabolic pathways. The NAD^+^/NADH ratio is a crucial regulator of cellular redox balance, influencing key biochemical processes, including fermentation and the biosynthesis of metabolites like succinate. Glycolysis results in a net gain of +2 NADH per glucose molecule, while the oxidative phase of the pentose phosphate pathway generates +1 NADPH. In contrast, the C4 metabolic pathway leading to succinate production consumes −2 NADH. Meanwhile, the C3 pathway producing acetate generates +1 NADH, whereas formate formation has a negligible NADH balance change (+/− 0 NADH). Ethanol production consumes two equivalents of NADH per mole of product, and lactate formation consumes one equivalent of NADH. This balance provides insight into electron flow and energy distribution across metabolic pathways [15]. An overview of the NADH balance is presented in Table 1.

### 1.3. Electrobiotechnology

A promising strategy to increase the succinate yields of *A. succinogenes* is fermentation in bioelectrochemical systems (BESs). This approach combines the fields of biotechnology and electrochemistry, an interdisciplinary approach known as electrobiotechnology. This combination allows the key advantages of biological components, such as self-replication, whole-cell catalysts, and high reaction specificity, to be linked with electrochemical methods to enable efficient and sustainable processes [16]. BESs are primarily applied in three key areas: generating electricity in microbial fuel cells, to produce hydrogen or methane in microbial electrolysis cells, and to produce more complex products such as butanol, succinate, or terpenes as part of electrochemically driven or assisted microbial electrosynthesis (MES) [17,18,19,20,21]. The term MES can either be defined broadly and include all microbially catalyzed electrochemical reactions that convert substrates into desired products, taking into account both anodic and cathodic processes, or it can be subdivided into the terms MES and electro-fermentation [22]. In the narrower sense, MES exclusively describes microbially catalyzed electrochemical processes that are based on the reduction of CO_2_ and in which electrons and the electric current act as the driving force. In contrast, the term electro-fermentation refers to self-sufficient fermentation processes on organic carbon compounds. Here, electrons are used to manipulate the metabolism by influencing the intracellular redox potential. In this context, the electric current does not act as a target product or primary energy source, but rather as a stimulator that affects the fermentation process by triggering the metabolism [23].

BES processes are typically performed in two-chamber H-cell reactors in which the anode and cathode chambers are separated [24]. Simplified single-chamber systems are an alternative variant. Both approaches offer specific advantages and disadvantages. In two-chamber systems, higher internal electric resistances can occur due to the membrane as well as pH gradients. In contrast, single-chamber systems can be negatively influenced by side reactions at the non-separated counter electrode, whereby the formation of oxygen is a particular concern [25,26].

### 1.4. Electro-Fermentation with A. succinogenes

Fermentation in the BES with *A. succinogenes* is an innovative method for optimizing biotechnological processes using electrical currents and increases the yield of target metabolites such as succinate. This is achieved primarily by shifting the NAD^+^/NADH ratio in the direction of NADH. The two mechanisms by which the transfer of electrons could theoretically occur are the direct transfer of electrons from the cathode to the organisms without a mediator or the transfer of electrons to a mediator, which then passes the electrons on to the organism [23]. In a 1999 study, Park and Zeikus [18] utilized an electrochemical bioreactor system with a working volume of 300 mL. This two-chamber system, separated by a Nafion membrane, comprised an anode and cathode compartment. 100 µM neutral red (NR) was used as a redox mediator. A potential of +2.0 V was applied between the electrodes, corresponding to approximately +2.2 V vs. NHE, with an Ag/AgCl electrode used as reference. The study’s findings revealed that in the absence of electrical energy, 47.33 mM glucose was consumed, while consumption increased to 60.44 mM with the presence of electrical energy. Regarding succinate formation, the absence of electrical energy resulted in 51.27 mM succinate, whereas electrical energy led to an increase to 82.88 mM, which corresponds to an increase of 20% [18]. In a subsequent study, Peng et al. (2024) [27] used a redox potential-controlled microbial electrolysis system to improve succinic acid production. The reactor system consisted of a two-chamber system with each chamber having a volume of 280 mL. By applying a continuous low potential of −400 mV to −600 mV with an Ag/AgCl electrode and adding neutral red as an electron mediator, an increase of 13% over the control was measured [27]. In a previously published paper of our group, there was an increase of 33% in the product titer compared to the reference operated at a potential of −600 mV relative to an Ag/AgCl electrode in combination with carbon nanotubes. The experimental setup comprised a 2-L bioreactor operating within a single-chamber system. The cathode used was a carbon fabric electrode from Kynol Europa GmbH. A platinum wire was utilized as the counter electrode [28]. Furthermore, the application of an electrical potential in a BES was shown to improve the intracellular NAD^+^/NADH ratio, leading to an increase in succinate yield compared to traditional fermentation methods [29]. The choice of electrode material also plays a decisive role. Properties such as the surface area and porosity can influence the adhesion of microorganisms and substrate diffusion [30].

While previous work described the potential of electrochemically assisted fermentation with *A. succinogenes* (e.g., Park & Zeikus, 1999) and investigated solid electrode potentials with conventional materials (Peng et al., 2024), the present study investigates the targeted use of low-cost carbon fabric electrodes as an alternative to precious metal-based systems [18,27]. In addition, it systematically analyzes the extent to which applied potentials close to the water electrolysis threshold can lead to the in situ formation of hydrogen in order to specifically promote the reductive metabolism of *A. succinogenes.* In this context, the aim of the study reported here was to analyze the influence of different electric potentials on the fermentation of *A. succinogenes* in the BES. The focus was on analyzing the effects of applied potential on product formation of succinate, acetate, formate, and lactate as well as on the effects of side reactions at the electrodes, especially hydrogen and oxygen formation. To understand the presumed effects of different potentials in more detail, further investigations were performed focusing on the properties of the electrode materials and their influence on by-product formation, biomass, and biofilm formation as well as on proteomic changes.

## 2. Materials and Methods

### 2.1. Microorganisms and Growth Conditions

*A. succinogenes* DSM 22257 strain 130Z was ordered from the Leibniz Institute DSMZ German Collection of Microorganisms and Cell Cultures (Braunschweig, Germany). Initially, the medium for the precultures was prepared with tryptic soy broth (TSB) [31]. Following medium preparation, the pH level was adjusted to 6.8. Subsequently, the medium was autoclaved. For the primary fermentation, the medium of Wang et al. (2018) [32] was used: 30 g·L^−1^ glucose, 31.5 g·L^−1^ Na_2_HPO_4_·12H_2_O, 10 g·L^−1^ NaHCO_3_, 8.5 g·L^−1^ NaH_2_PO_4_, and 5 g·L^−1^ yeast extract. Prior to inoculation, the medium underwent autoclaving, with the glucose being autoclaved separately from the other medium components. Initially, precultures were prepared with the TSB medium. For this purpose, 120 mL flasks were filled with the medium and subsequently degassed with N_2_ for 20 min to establish anaerobic conditions. Subsequently, *A. succinogenes* was inoculated and incubated for 16 h at 37 °C and 80 rpm.

### 2.2. Biofilm Analysis

Serum bottles with a working volume of 50 mL were prepared to investigate the formation of biofilms on various carbon fabrics from Kynol Europe GmbH, Hamburg, Germany. The carbon fabrics were first dried at 120 °C for 2 h, then dehydrated for 30 min in a desiccator, then weighed, and then placed in the serum bottles and autoclaved. Each carbon fabric specimen exhibited dimensions of 2 cm × 1.5 cm. The carbon fabrics with the designations ACC-5092-10, -15, -20, and -25 were utilized in this study. The ACC-5092 series of activated carbon fibers exhibit high specific surface areas, with ACC-5092-10 possessing a surface area of 1200 m^2^·g^−1^, ACC-5092-15 reaching 1500 m^2^·g^−1^, and ACC-5092-20 demonstrating a surface area of 1750 m^2^·g^−1^. The fabric variant ACC-5092-25 has the highest surface area, measured at 2100 m^2^·g^−1^. These surface area measurements were determined by Kynol using nitrogen (N_2_) adsorption at 195 °C. The heights of the meshes were taken from the manufacturer’s specifications and are 0.66 mm for ACC-5092-10, 0.6 mm for -15, 0.55 mm for -20, and 0.54 mm for -25. Subsequently, the primary fermentation medium was transferred to the serum bottles. The serum bottles were then sealed with rubber butyl stoppers and crimped. Subsequently, two cannulas, one of which had a length of 15 cm and another a length of 5 cm, were inserted through the stoppers. The stoppers were then subjected to a process of degassing, which involved the utilization of nitrogen (N_2_) for a duration of 20 min and carbon dioxide (CO_2_) for a duration of 10 min. Following this, the inoculation and incubation steps were initiated. The bottles were subjected to an incubation period of 48 h with shaking at 80 rpm and at a temperature of 37 °C. Subsequently, the carbon fabrics were removed from the bottles and then were subjected to a drying process at a temperature of 120 °C in a drying oven for a duration of 18 h and then in a desiccator for 30 min. Following this, the carbon fabrics were weighed to ascertain the amount of weight gain. The microscopic analysis of the biofilms was performed with a Leica SP5 II confocal laser scanning microscope (CLSM) (Leica Microsystems GmbH, Wetzlar, Germany) using a HC PL Fluotar 10x/0.30 objective (dry). A 405 nm UV diode with 50% intensity and a 488 nm argon laser with 15% intensity were used. Images were captured using the PMT3 (500–520 nm, green) and PMT5 (600–660 nm, red) detectors. The biofilm samples were first cultivated in a bioelectrochemical system on a carbon fabric electrode (ACC-5092-15) and then stained with fluorescein diacetate (FDA) and propidium iodide (PI). FDA was dissolved at a concentration of 0.7 mg∙ml^−1^ in acetone and PI at a concentration of 1 mg∙mL^−1^ in phosphate buffered saline (PBS, pH 8). For staining, the biofilms were transferred to 5 mL phosphate buffer (pH 8), mixed with 200 µL PI solution and 125 µL FDA solution, and incubated for 30 min.

### 2.3. Carbon Fabric Electrode Analysis

To better understand the fabrics for use as electrodes in the BES, it was important to understand the electrical properties of these materials. Therefore, they were examined regarding their electrical resistance. For this purpose, the fabrics ACC-5092-10, -15, -20, and -25 from Kynol Europe GmbH were cut to 30 mm × 20 mm. A crocodile clip was then applied to both sides of the fabric and the electrical resistance was measured with a multimeter (Voltkraft M-4650B, Conrad Electronic SE, Hirschau, Germany). An experiment was conducted to analyze the hydrogen formation at the electrodes as a function of applied potential. In this experiment, bottles with a liquid volume of 250 mL were used. The overhead space was 50 mL. These bottles were then sealed with butyl rubber stoppers. Two cannulas and two platinum wires were inserted through each bottle. Subsequently, a 4.5 cm × 2.0 cm piece of ACC-5092-15 carbon fabric was affixed to the terminal end of the platinum wire, and these were the sole components in contact with the medium. Additionally, an Ag/AgCl reference electrode (SE11NSK7-4 from Meinsberg Sensortechnik, Xylem Analytics Germany Sales GmbH & Co. KG, Weilheim, Germany) was inserted through the butyl catheter plug. Subsequently, 250 mL of Wang medium was introduced into the bottles, and the entire preparation was degassed for 20 min using N_2_ and 10 min using CO_2_. The electrodes were then connected to the potentiostat (MultiPalmSens4 from PalmSens, Houten, The Netherlands), and the corresponding potential was set. The entirety of the experiment was conducted at a temperature of 37 °C, with stirring. The hydrogen formed in the headspace of the bottles was measured using a GC AK LCGC 15, developed by Kappenberg (Münster, Germany) [33].

### 2.4. Bioelectrochemical Setup

Benchtop-scale experiments with a reactor volume of 130 mL were performed in a single-chamber BES with an integrated three-electrode setup. The working electrode (WE) and the counter electrode (CE), with a geometric surface area of 17.5 cm^2^, limited by the size of the reactor chamber, were made of activated carbon fiber (ACC5092-15, KYNOL EUROPA GmbH, Hamburg, Germany) and were connected to the potentiostat MultiEm Stat3 (PalmSens, Houten, The Netherlands) via platinum wires (Ø = 0.4 mm). An Ag/AgCl electrode with saturated KCl solution (Sensortechnik Meinsberg, Waldheim, Germany) served as a reference electrode. During fermentation, the pH value was maintained above 6.8 by adding 5 M NaOH, regulated by a Profilux 3.1T computer system (GHL, Kaiserslautern, Germany). Temperature control and mixing were achieved using a heating and stirring plate (MULTI-HS 6, Carl Roth GmbH, Karlsruhe, Germany).

### 2.5. Fermentation in Parallel Bioreactor

Experiments were also carried out in controlled batch reactors as a triple determination. The objective of this investigation was to examine the impact of elevating the oxygen content of the gassing on the metabolic activity of the organisms. CO_2_ was utilized as the secondary gas for the purpose of mixture formation. Fermentation was conducted in parallel in three 500 mL bottles, each with a fermentation volume of 400 mL. The bottles were sealed with black GL 45 butyl rubber stoppers after the addition of a magnetic stirring bar. Two cannulas (one long and one short) were inserted through the stopper. These cannulas were utilized for two primary functions: the first was to facilitate the process of sampling, and the second was to regulate the pH level. A pH electrode (Hamilton Bonaduz AG, Bonaduz, Switzerland) was inserted into one of the bottles and connected to the bioprocess controller (BioFlo 120 from Eppendorf SE, Hamburg, Germany). The pH was regulated with 5 M NaOH. Since all the bottles were configured as parallel fermenters, one was designated as a reference for the pH value, while the other two bottles were regulated in parallel via the bioprocess controller and manually checked by sampling. Everything was autoclaved before fermentation. A dissolved oxygen sensor DO-BTA (Vernier, Beaverton, OR, USA) was inserted into one of the bottles. Prior to insertion, the bottle was meticulously cleansed with Bacillol AF^®^, as it could not be autoclaved. This sensor was utilized to quantify the dissolved oxygen concentration throughout the fermentation process. To ensure uniform mixing and temperature control within the vials, the vials and reactors were collectively incubated on the six-position Multi-HS 6/15 hotplate from (Velp Scientifica SrL, Usmate Velate, Italy). Mixing in the reactors was carried out at 400 rpm using a stirring bar.

### 2.6. Proteome Analysis

The biomass samples from the electrode surface were lyophilized, stored at −80 °C, and resuspended in a urea solution (8 M) before being sonicated. The extracted proteins were treated with 20 mM dithiothreitol and 5 µL of a 475 mM iodoacetamide solution, followed by enzymatic digestion with 5 µg trypsin overnight. Finally, the peptides were purified, dried, resuspended, and analyzed by LC-MS/MS. The proteome analysis was performed by the Analytical Resources Core of Colorado State University (USA). For each sample, 1 µg of peptides was purified and concentrated, then fractionated using reverse-phase chromatography with a solvent gradient between 0.1% formic acid in water (A) and 80% acetonitrile with 0.1% formic acid (B). Peptides were eluted directly into the mass spectrometer (Orbitrap Eclipse, Thermo Scientific, Waltham, MA, USA) equipped with a Nanospray Flex ion source (Thermo Scientific, Waltham, MA, USA). Proteome Discoverer (PD) 3.0 was used for data processing (Thermo Scientific, Waltham, MA, USA). Raw data were interrogated against the FASTA file of the reference proteome for *Actinobacillus succinogenes* (strain ATCC 55618/DSM 22257/CCUG 43843/130Z) from Uniprot. Thresholds were set such that the false discovery rate (FDR) was ≤1%, and protein identification was defined as the presence of at least one peptide. More information about sample preparation, analysis, and data processing can be found in the Appendix A.

### 2.7. Substrate and Product Analysis

The liquid-phase fermentation products were analyzed by liquid chromatography (HPLC), as described in previous publications [28,29,33]. The measurements were carried out at 30 °C and 5 mM sulfuric acid mobile phase, but due to difficult resolution of the lactate and succinate peaks, additional analyses were performed at 80 °C using a 2.5 mM sulfuric acid concentration. The detector utilized in this experiment was a refractive index detector (Infinity II Refractive Index Detector from Agilent Technologies Inc., Santa Clara, CA, USA). The analysis of gas samples from the headspace of serum and Schott bottles was performed using the Mini GC AK LCGC 15 system, developed by Arbeitskreis Kappenberg (Münster, Germany). The chromatographic separation was carried out using a Chromosorb 102 column from Arbeitskreis Kappenberg (Münster, Germany) with a length of 0.8 m. Air was employed as the carrier gas. Detection was achieved with a thermal conductivity detector (TCD) from Arbeitskreis Kappenberg (Münster, Germany). Column temperature was maintained at 20 °C to ensure optimal separation efficiency. The injection volume of the sample was set at 1 mL. Total analysis time ranged from 90 to 120 s, which was sufficient to fully resolve the gaseous components, including H_2_ [33].

## 3. Results and Discussion

### 3.1. Biofilm Analysis

A critical aspect was the detailed analysis of the carbon fabric electrode materials used in the BES system. Four different activated carbon fabrics, provided by Kynol Europe GmbH, were investigated to evaluate their suitability for biofilm formation and electron transfer processes. The fabrics tested, ACC-5092-10, ACC-5092-15, ACC-5092-20, and ACC-5092-25, varied in their surface properties according to Kynol, which could affect their performance in biofilm attachment and overall bioprocess efficiency. Activated carbon electrodes are produced by carbonization of precursor materials, followed by activation processes that increase their porosity and surface area [34]. The resulting porous structure improves the electrochemical performance of these electrodes. The increased surface area and tailored porosity resulting from activation processes are crucial for optimizing the performance of activated carbon electrodes in various electrochemical applications [35]. The primary objective was to evaluate the ability of the electrodes to support biofilm formation, a critical factor for efficient electron transfer in BESs. To this end, biomass accumulation on the fabrics was quantified by measuring weight gain after incubation with microbial cultures. A key challenge was to distinguish between microbial biomass and medium components that might bind non-specifically to the fabrics. To address this, three control experiments were performed for each fabric type, resulting in n = 3 replicates for all conditions. Measurements were corrected using the controls, and standard deviations were calculated using error propagation methods. The results show a clear trend in biofilm formation across the different fabric types (Figure 1). As shown in Figure 1B, the increase in biomass for ACC-5092-10 was 3.5 ± 0.2%. For ACC-5092-15, the increase was 5.1 ± 0.4%, for ACC-5092-20, 5.7 ± 0.3%, and for ACC-5092-25 the largest increase of 6.8 ± 0.5% was measured. This is consistent with surface activation. Figure 1A also shows a microscopic image taken using CLSM. The illustration shows the ACC-5092-15 fabric, highlighting biofilm formation on the material’s surface and the integration of the biofilm into the fibers. The biofilm, which fluoresces green due to metabolized fluorescein diacetate, indicates that most organisms within the biofilm remain metabolically active after 48 h of fermentation. These results correlate strongly with the surface activation times of the fabrics as detailed in the technical specifications provided by Kynol Europe GmbH. Longer activation times appear to enhance surface properties, particularly roughness, favorable for biofilm adhesion. This is consistent with studies showing that hydrophobic surfaces with high surface roughness promote biofilm formation [36]. Activation can optimize the pore structure of the carbon material, resulting in an increased active surface area. This provides more adhesion sites for microorganisms and promotes mass transfer [37]. In addition, activation can generate functional groups on the surface that favor electron transfer [38]. The biofilm formation and the higher activation can have a negative effect on the performance of the BES. In addition, the biofilm can increase the internal resistance of the electrode, and the conductivity of the electrode can be reduced by excessive activation [37,39].

### 3.2. Carbon Fabric Electrode Analysis

The electrical conductivity of the various carbon fabrics was measured (Figure 2A). Rectangular samples with dimensions of 20 mm × 15 mm, 20 mm × 30 mm, and 20 mm × 45 mm (length × width), were measured to ascertain electrical resistance. The measurements were conducted along the longer side (i.e., across 15, 30, and 45 mm, respectively). The measured electrical resistance is shown in Figure 2A. The electrical resistance of the fabric increased with increasing activation. With a dimension of 20 mm × 45 mm, a resistance for ACC-5092-10 of 24.4 ± 0.5 Ω, for ACC-5092-15 of 28.6 ± 0.5 Ω, for ACC-5092-20 of 45.4 ± 1.4 Ω, and for ACC-5092-25 of 110.7 ± 8.9 Ω was measured. Based on the conductivity, the resistivity could be determined by the height of the carbon fabric. The thicknesses of the fabrics are 0.66 mm for ACC-5092-10, 0.6 mm for -15, 0.55 mm for -20, and 0.54 mm for -25. The specifications in Figure 2B refer to the dimensions of 20 mm × 45 mm. In contrast to 20 mm × 15 mm, for other dimensions the resistivity calculations differed slightly. One reason for this is measurement deviations caused by the small size, which could result from slightly shifting the electrodes for the measurement. This resulted in resistivity values of 7.1 × 10^−3^ Ω·m for ACC-5092-10, 7.6 × 10^−3^ Ω·m for ACC-5092-15, 1.1 × 10^−2^ Ω·m for ACC-5092-20, and 2.7 × 10^−2^ Ω·m for ACC-5092-25. The lowest measured value of 7.1 × 10^−3^ Ω·m, for ACC-5092-10, is above the resistivity of activated carbon. This is in the range of 0.2 Ω·cm, which corresponds to 2 × 10^−3^ Ω·m. The resistivity of graphite is 1.3 × 10^−5^ Ω·m, for titanium 4.2 × 10^-7^ Ω·m [37,40,41], for copper, 1.7 × 10^−8^ Ω·m, and for silver, 1.6 × 10^−8^ Ω·m [42]. In a study by Dumas et al., it was shown that carbon felt has a high porosity for microbial adhesion [43,44]. There are also statements that electrons cannot be transferred directly between cells and electrodes, and that biofilms on electrode surfaces can therefore restrict electron and mass transfer due to the increasing resistance and the resulting voltage drop [44,45]. For this reason, the ACC-5092-15 fiber was chosen for BES, as the advantage of biofilm formation could still be combined with the lower specific resistance.

Since the biofilm formation on ACC-5092-15 was good but the resistance was lower than with ACC-5092-20, further fermentations were conducted with this electrode. It was therefore of interest to investigate the reactions on the electrode in the form of water electrolysis. This investigation provides important findings for electro-fermentation with *A. succinogenes*. Hydrogen was measurable after 240 min in the overhead compartment at applied potentials of −1333 mV, −1666 mV, and −2000 mV with an Ag/AgCl reference electrode. At −1000 mV, no hydrogen formation was observed. These results are consistent with the theory of water electrolysis. Due to overvoltage at the electrodes, water electrolysis only starts from approx. −1500 to −1800 mV, depending on the electrode materials used, the electrolyte, and the operating conditions [46]. It was also found that there was a drop in pressure formed in the overhead space in some bottles. This is due to the CO_2_ used for gassing, which was still in the overhead space and was then metabolized during fermentation. As can be seen in Figure 3A, the H_2_ content in the overhead space increases by 8.23% for −2000 mV, 2.69% for −1666 mV, and 0.92% for −1333 mV. Based on the ideal gas law, the amount of substance in the overhead space shown in Figure 3B was then determined. Here, at a potential of −2000 mV, a substance quantity of 0.163 ± 0.062 mmol could be measured, which represents the highest value of H_2_. Assuming a stoichiometric H_2_:O_2_ ratio of 2:1 in water electrolysis, the formation of 0.16 mmol H_2_ corresponds to the simultaneous generation of 0.08 mmol O_2_ within 240 min [47]. In addition, the electric charge was also plotted in Figure 3B. At an applied potential of −2000 mV, 54.81 ± 2.28 C (Coulombs) entered the system. This corresponds to a Faraday efficiency of 57%. In contrast, at −1333 mV, the amount of substance was only 0.016 ± 0.008 mmol, with a Faraday efficiency of 13%. At −2000 mV, the electrode surface, which had an area of 18 cm^−2^ considering both active sides of the electrode, had an electrode efficiency of 0.009 mmol·cm^−2^. Hydrogen plays a central role as an electron donor in microbial systems. Previous studies, such as that of Park and Zeikus (1999), showed that electrochemically generated hydrogen and the use of neutral red as a mediator can increase the availability of NADH, which promotes the reduction of CO_2_ to succinate [18]. Our results confirm this possibility and show that hydrogen can be generated directly at the electrode, eliminating the need for external hydrogen sources. This provides a direct way to manipulate the redox potential and control metabolism in *A. succinogenes*. It could therefore make sense to produce H_2_ at the electrode. In a study by McKinlay et al. (2008) [48], the influence of H_2_ on the metabolism of *A. succinogenes* was investigated using 13C flux analysis. *A. succinogenes* reacts to high CO_2_ and H_2_ concentrations by producing more succinate and less formate, acetate, and ethanol [48].

### 3.3. Effect of Different Potentials on Product Formation

Apart from investigating the influence of different carbon fabrics on the fermentation process in BESs, it is also of major interest how different electric potentials affect the fermentation and metabolism of *A. succinogenes*. The previously selected carbon fabric ACC-5092-15 was used, as it combines favorable properties for biofilm formation with a low specific resistance.

Six different electric potentials were selected to investigate their influence on the fermentation of *A. succinogenes* in a single-chamber system specially designed for cathodic electro-fermentation with a laboratory scale of 130 mL. After 72 h, the final samples were analyzed for their product composition (Figure 4A). The results of the control experiments conducted at 0 mV illustrate the typical product distribution associated with increased CO_2_ availability. Succinate was the primary product, reaching a concentration of 18.37 ± 0.55 g∙L^−1^. In contrast, lactate reached only 0.59 ± 0.12 g∙L^−1^ at the end of the process. Among the tested potentials, −500 mV and −1000 mV yielded the highest succinate titers, with −500 mV standing out at 19.76 ± 0.41 g∙L^−1^. This observation aligns with previous studies reporting a positive impact of potentials between −600 and −800 mV on succinate production [32,49]. Production of the by-products acetate and formate was consistent at potentials up to −1000 mV. A marked increase, particularly in acetate titer, is observed at potentials starting from −1333 mV. Consequently, the final acetate titer at −2000 mV exceeds that of the control by 1.48 ± 0.14 g∙L^−1^. A more pronounced effect is noted for succinate; while moderate potentials between −1000 and −500 mV enhance succinate production, lower potentials negatively impact the process. At −2000 mV, the overall metabolic activity is inhibited, as indicated by a residual glucose concentration of 15.62 ± 0.74 g∙L^−1^ after 72 h. Additionally, a substantial increase in lactate production is observed at lower potentials. The highest lactate titer of 3.12 ± 0.21 g∙L^−1^ is recorded at −1666 mV, representing a production increase of up to 5.65 ± 0.44%∙10^2^ compared to the lower potentials or the control. A direct comparison reveals that while lactate constitutes only 2% of the total product distribution at −1000 mV, this proportion rises to 12% at −1666 mV. Correspondingly, the proportion of succinate decreases from 70% to 55%.

Based on the results of these experiments, the applied potential has an influence on the metabolism of *A. succinogenes*. Regarding the goal of increased succinate production, more negative potentials were associated with a decrease in succinate production. What could cause this phenomenon? Given the fact that the cathode and anode chambers were not separated in the experimental setup, it can be assumed that reactions at both electrodes at lower potentials trigger the observed product shift [16,50]. Particularly notable here is the formation of hydrogen and the associated formation of oxygen, which has already been demonstrated in the abiotic experiments for carbon electrode materials. The influence of oxygen on the metabolism of *A. succinogenes* has already been addressed in previous studies, which have also shown that the relatively low lactate production of *A. succinogenes* can be enhanced greatly by two-phase fermentation. In this process, the organism is first cultured in an aerobic phase before switching to an anaerobic phase with CO_2_ aeration [51]. According to the literature, this process regulation has led to an up to 32-fold increase in lactate levels. Here, the initial oxygen supply in the aerobic phase should promote cell growth and support the shift of the C4 to C3 flux. This is also reflected in an up to 18-fold increase in the concentration of the enzyme lactate dehydrogenase in the two-phase culture compared to a purely anaerobic monophase culture [52]. The observations by Li et al. (2010) on the influence of oxygen on metabolism could perhaps explain the phenomena observed in this study in the single-chamber BES [51]. The oxygen produced at the anode through electrolysis might cause a kind of two-phase fermentation. This would cause a shift towards C3 metabolism, which could explain the decrease in succinate production and the simultaneous increase in the production of lactate and acetate. However, some aspects contradict this first explanation.

The described influence of oxygen does not explain the negative effect on general product formation, cell growth, and the associated decrease in biomass, as observed at a potential of −2000 mV. The distribution of the cell dry mass between the working electrode and the remaining reactor is comparable in the control and at potentials between −500 and −1333 mV. A decrease in biomass is only observed at lower potentials, which is particularly visible at the WE (see Figure 5). As *A. succinogenes* is facultatively anaerobic, oxygen production should not have a negative effect on biomass [53]. On the contrary, increased biomass formation would be expected under aerobic conditions. The inhibition observed here is therefore probably not caused by oxygen but is likely caused by electrochemical side reactions with the medium [16]. These reactions might not be caused by the simplified single-chamber system. Experiments in two-chamber BESs with *A. succinogenes* show similar inhibitory effects on growth and the formation of succinate and acetate at potentials above −2600 mV. The difference of −600 mV compared to the −2000 mV investigated here is related to the specific properties of the BESs, in particular the electrode materials used and the internal resistances of the systems. Unfortunately, the change in lactate concentration at more negative potentials in two-chamber systems was not investigated in the study of Zhao et al. (2016), which means that no direct comparison can be made regarding oxygen development and its influence [49].

Furthermore, a pure second anaerobic phase should not occur in BESs, as the electric potential is applied during the entire fermentation process. Electrolysis and the associated oxygen formation should therefore occur throughout the entire process, which is not in line with the two-phase fermentation described by Li et al. (2010) [51].

Contrary to prevailing assumptions in the literature, our results indicate that oxygen generation at the anode does not influence lactate formation in *A. succinogenes*. The results of Li et al. (2010) suggesting that co-fermentation promotes lactate formation could not be verified under our experimental conditions [51]. To test these claims, a series of parallel batch reactor experiments were conducted using different aeration rates with a controlled mixture of CO_2_ and synthetic air. These optimized conditions conclusively demonstrated the absence of lactate formation across all aeration regimes. The results are summarized in Figure 6, with data obtained at 30 °C and 5 mM H_2_SO_4_ conditions included for consistency. Aeration was performed with varying proportions of compressed air, ranging from 5% to 100% of the gas stream, with the remainder being CO_2_. At 5% compressed air (95% CO_2_), a succinate yield of 0.71 ± 0.12 g_succinate_·g_glucose_^−1^ was obtained, accompanied by a by-product yield of 1.59 ± 0.31 g_succinate_·g_glucose_^−1^. When the amount of compressed air was increased to 10%, the succinate yield decreased to 0.57 ± 0.14 g_succinate_·g_glucose_^−1^ and the by-product formation decreased to 1.09 ± 0.27 g_succinate_·g_glucose_^−1^. Interestingly, at 50% air, an increase in succinate yield to 0.64 ± 0.17 g_succinate_·g_glucose_^−1^ was observed, although by-product formation remained suppressed at 1.02 ± 0.26 g_succinate_·g_glucose_^−1^. At 100% compressed air, corresponding to approximately 20% oxygen in the system, the succinate yield decreased to 0.36 ± 0.14 g_succinate_·g_glucose_^−1^, with by-product formation at 0.73 ± 0.39 g_succinate_·g_glucose_^−1^. These results challenge previous claims that oxygen generated at the anode or introduced by aeration affects product formation. The proportion of oxygen in the air is approx. 20.8% [54]. Even under conditions of 100% compressed air, succinate and by-product formation were not enhanced, suggesting that oxygen does not play a role in lactate production. Consequently, these results rule out oxygen generation by electrolysis as a contributing factor at elevated voltages. Instead, alternative effects, possibly related to redox balance or metabolic flux, are likely to influence the observed metabolic shifts. Aeration also changed the relative proportions of fermentation products. At 5% aeration, succinate was 59.6% of the total products. With increasing aeration, succinate decreased to 39.7% at 100% aeration. At the same time, formate increased from 15.1% to 23.5%, and ethanol production increased from 1.1% at 5% air to 11.5% at 100% air. These shifts in product distribution highlight the complex interplay between oxygen availability and metabolic pathways in *A. succinogenes.* The experiments were performed in parallel 400 mL bioreactors. Variability in the results, reflected by higher standard deviations, can be attributed to the experimental design and the challenges associated with maintaining consistent conditions across reactors. Standard deviations were calculated using error propagation methods to account for the observed variability. Contrary to the widespread belief that *A. succinogenes* is capable of producing lactate [55], no lactate formation was detected in the experiments with oxygen gassing. This observation is consistent with earlier reports by McKinlay et al. (2010), who also found no lactate production in *A. succinogenes* [6]. In a study by Guarnieri et al. (2017) [56], lactate formation was observed following targeted metabolic engineering, in which competing carbon fluxes towards succinate were eliminated, thereby revealing alternative by-product pathways. The resulting genetically engineered strains also provide information about the energy and redox balance as well as the mechanisms of organic acid biosynthesis [56]. The shift in carbon flux towards lactate under altered redox conditions may be explained by the need to maintain intracellular redox balance. Lactate production could serve as a mechanism to consume excess reduction equivalents to ensure redox homeostasis, and the pathway from pyruvate to lactate is shorter than to succinate. A recent study on the deletion of the *pflA* gene encoding the pyruvate formate lyase 1-activating protein showed interesting metabolic shifts. Under aerobic conditions, the deletion led to increased acetate accumulation, while under anaerobic conditions a complete conversion of pyruvate to lactate was observed [55]. The altered redox ratio induced by the applied potential could be a decisive factor for the activation of lactate production. A study by Li et al. (2010) showed that Ldh activity was almost 18-fold higher in a dual-phase fermentation process than in a single-phase process [51]. Thus, it could be that the oxygen gassing had no influence, but the increased reducing potential led to the formation of more reduction equivalents.

### 3.4. Proteome Analysis of Key Enzymes of C4 and C3 Metabolism

For a comprehensive analysis and physiological insights into the product titer shift of *A. succinogenes* induced by more negative electric potentials in BESs, key enzymes involved in the C4 and C3 metabolism were detected through proteome analysis (Figure 7). Based on the previous results, an electro-fermentation using the carbon fiber material ACC-5092-15 at −1000 mV was compared with a cultivation at −1500 mV to specifically assess the impact of water electrolysis. These investigations were conducted both in the single-chamber systems employed in this study and in two-chamber H-cell reactors under identical conditions (Table 2). The proteome data from the experiments in the single-chamber system revealed that the different experimental parameters do not cause any relevant changes in C4 metabolism, especially in the CO_2_-dependent reductive metabolic pathway of the TCA cycle, described by McKinlay and Vieille (2008) [48]. The enzymes phosphoenolpyruvate carboxykinase (PEPCK), malate dehydrogenase (Mdh), fumarate hydratase (Fm), and the subunits of fumarate reductase (Fr) show only slight differences between the experimental conditions. The results for PEPCK, which is present with a ratio of normalized abundances of 1.08, are particularly noteworthy. Since this enzyme catalyzes the conversion of phosphoenolpyruvate to oxaloacetate, it has an influence on the flux into the C4 metabolism and thus on succinate production. Previous bioelectrochemical studies with *A. succinogenes* have shown that PEPCK is upregulated in a fermentation with applied electric potential compared to control cultivation without potential, which could explain the increased succinate production in BESs [57]. The small change in the abundance ratio of the C4 enzymes in the present study indicates that the decrease in succinate concentration shown in Figure 4 is not caused by reduced enzyme levels of the reductive TCA cycle.

Similar results were obtained for the metabolic enzymes involved in the formation of the by-products acetate and formate. The normalized abundances of phosphate acetyltransferase (Pc) and acetate kinase (Ak), which are involved in acetate production, differed by a maximum of 0.09 between applied potential and control experiments for both reactor types. The results demonstrate that only minor fluctuations were observed in the normalized abundances of the enzymes involved in by-product formation, which explains the slight change in by-product content from 13% to 14% formate and from 15% to 19% acetate. A clear change can be observed in lactate dehydrogenase (Ldh) Asuc_0005. For electro-fermentation in the single-chamber system, the normalized abundance was 2.93-fold higher at −1500 mV compared to −1000 mV. These results are consistent with the increase in lactate content in product distribution of 10% (Figure 4). This observation is in line with the data from Li et al. (2010), who also demonstrated increased Ldh expression [51]. Despite these changes, it is not a two-phase fermentation due to the continuous electrolysis. In addition, the oxygen gassing experiments in this study indicate that oxygen concentrations are not the main cause of the observed increase in lactate and decrease in succinate in BESs. The proteome analyses of the two-chamber experiments under comparable conditions (Table 2), support this thesis. In addition to the absence of major changes in C4 metabolism and in the by-products acetate and formate, the normalized abundance of Ldh for lactate production increased almost threefold in the two-chamber setup. Separation into two chambers prevents oxygen from being produced as a by-product in the fermentation chamber during cathodic electro-fermentation. It is known that semipermeable membranes such as Nafion 117 allow a small amount of oxygen permeation [58]. Yet this effect is so minor that it cannot be compared with direct gas generation in the cathode chamber. It is therefore very likely that it is not primarily oxygen, but rather the lower electric potential and by-products such as hydrogen that cause the observed product shifts of *A. succinogenes*. Studies, like the one by Park and Zeikus (1999), show that hydrogen can increase the availability of NADH under certain conditions, which influences the intracellular redox potential of the microorganisms [18]. In addition, electro-fermentations with an applied potential can target a controlled stimulation of the NADH pool. An applied potential of −800 mV led to an increase in the NAD^+^/NADH ratio towards NADH, particularly in the stationary phase of fermentation [29]. The metabolic shift towards lactate under these altered redox conditions can therefore probably be attributed to the organism attempting to maintain the intracellular redox balance. Excess reduction equivalents are metabolized via lactate formation, which could explain the increased production and could also contribute to redox homeostasis.

## 4. Conclusions and Outlook

Overall, this study showed that different electric potentials affect the metabolism and consequently the product distribution of *A. succinogenes* in bioelectrochemical systems. The specific effects are strongly dependent on the system used, especially on the electrode material, which plays a crucial role. Different carbon fabric electrodes influence both the biofilm formation in the reactor and the specific resistance of the system. This in turn affects the metabolism, depending on the applied potentials, which is caused by side product formation at the electrodes. More negative potentials lead to a decrease in succinate titers and an increase in by-product formation, especially lactate, which is reflected in the 2.93-fold increase in the normalized abundance of Ldh. It had been assumed that this effect is directly related to electrolytic processes at the electrodes and the resulting formation of hydrogen and oxygen. However, this study demonstrated that the oxygen produced at the electrode is not the only factor responsible for the formation of lactate; rather, it is a combination of several effects, including influence on the intracellular redox potential. These results show that lower (more reducing) electric potentials applied in an electro-fermentation process do not necessarily lead to an improved fermentation process. Instead, they can lead to an unintended change in product formation or inhibition of metabolism, especially due to the side reactions at the electrodes, as demonstrated with *A. succinogenes*. Future work should evaluate the stability, biofilm integration, and electrochemical efficiency of the carbon fabric electrodes in continuously operated long-term bioelectrosystems under real process conditions. In addition, the targeted genetic engineering of *A. succinogenes* could offer the possibility of suppressing competing C3 secondary pathways such as lactate formation and controlling the input of reduction equivalents in a targeted manner via electrochemically supplied electrons. The aim of this measure would be to further increase the succinic acid yield.

## Figures and Tables

**Figure 1 microorganisms-13-01720-f001:**
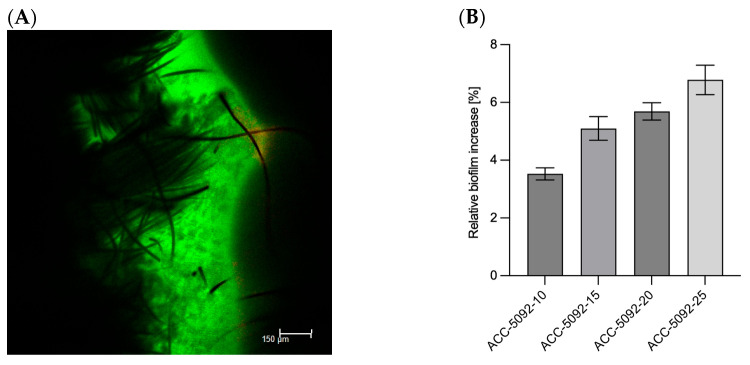
Analysis of biofilm formation on different carbon fabric electrodes after 48 h fermentation with *A. succinogenes*. Experimental parameters: anaerobic; main culture medium. V = 0.05 l; 80 rpm; T = 37 °C; initial pH = 6.8; n = 3. (**A**) Confocal laser scanning microscope image of the biofilm stained with fluorescein diacetate and propidium iodide. The black structure is the carbon fiber material, ACC-5092-15. The green structure is the biofilm. (**B**) Relative weight increase of the biofilm on the respective carbon fabrics.

**Figure 2 microorganisms-13-01720-f002:**
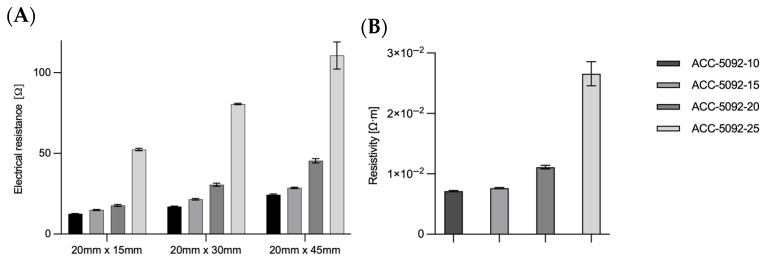
Analysis of the electrical resistance and resistivity of the carbon fabric. The resistance was measured over three different lengths with the same width. The carbon fabrics inherently have different thicknesses, from which the resistivity is derived. The thicknesses of the fabrics are 0.66 mm for ACC-5092-10, 0.6 mm for -15, 0.55 mm for -20, and 0.54 mm for -25. Experimental parameters: n = 3; sizes 20 mm × 15 mm, 20 mm × 30 mm, and 20 mm × 45 mm; resistance was measured with a multimeter (Voltkraft M-4650B, Conrad Electronic SE, Hirschau, Germany). (**A**) Plot of the measured electrical resistance for three different sizes of carbon fabric. (**B**) Resistivity based on the size 20 mm × 45 mm.

**Figure 3 microorganisms-13-01720-f003:**
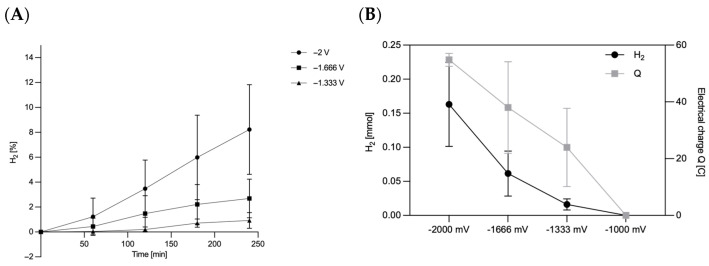
Analysis of hydrogen formation on the carbon fiber ACC-5092-15. Experimental parameters: anaerobic; main culture medium. V = 0.25 l; T = 37 °C; initial pH = 6.8; under stirring; n = 3. ACC-5092-15 with a size of 4.5 cm × 2 cm was affixed to the terminal end of the platinum wire and Ag/AgCl reference electrode, with the given potentials of MultiPalmSens4. (**A**) Percentage share of hydrogen in the overhead space over 240 min. (**B**) Yield of H_2_ after 240 min of reaction and the respective electrical charge.

**Figure 4 microorganisms-13-01720-f004:**
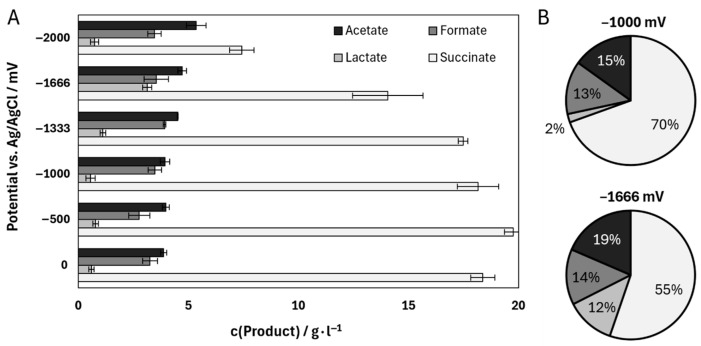
(**A**) Product concentrations of the main products of a fermentation of *Actinobacillus succinogenes* under different electric potentials (Ag/AgCl with saturated KCl). (**B**) Percentage product distribution of the four most important products of the *A. succinogenes* fermentation. The fermentations were performed in a single-chamber bioelectrochemical system with the following parameters: T = 37 °C; n = 2; 200 rpm; 130 mL; 30 g L^−1^ glucose; pH = 6.8, regulated with 5 M NaOH and gassing with 0.2 vvm CO_2_. The carbon fiber ACC-5092-15 was used as electrode material.

**Figure 5 microorganisms-13-01720-f005:**
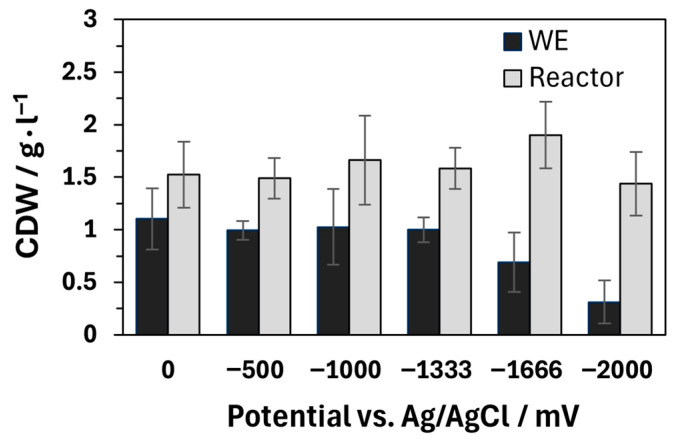
Cell dry weight (CDW) from the biofilm on the working electrode (WE), plus remaining CDW from the reactor. The fermentations were performed in a single-chamber bioelectrochemical system with the following parameters: T = 37 °C; n = 2; 200 rpm; 130 mL; 30 g L^−1^ glucose; pH = 6.8, regulated with 5 M NaOH and gassing with 0.2 vvm CO_2_. The carbon fiber ACC-5092-15 was used as electrode material.

**Figure 6 microorganisms-13-01720-f006:**
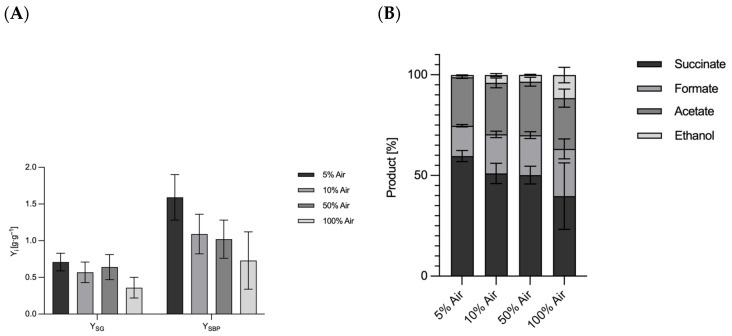
(**A**) Overview of the product and by-product yields and distributions of the fermentation of *A. succinogenes* after 48 h with different proportions of air in the fumigation in parallel fermentation. (**B**) Overview of product distribution of the fermentation of *A. succinogenes* after 48 h with different proportions of air in the fumigation in parallel fermentation. Experimental parameters: anaerobic; gassing 0.083 vvm (as indicated, part compressed air and the rest CO_2_), main culture medium. V = 0.4 l; 400 rpm; T = 37 °C; pH = 6.8; pH controlled with 5 M NaOH; n = 3. HPLC parameters: Repromer H^+^ column, 5 mM H_2_SO_4_; t = 30 min; flow rate 0.6 mL·min^−1^ at 30 °C. Lactate detection at 5 mM H_2_SO_4_ and 80 °C. (**A**): Plot of succinate yield and new product yield related to glucose used. (**B**): Plot of the product distribution.

**Figure 7 microorganisms-13-01720-f007:**
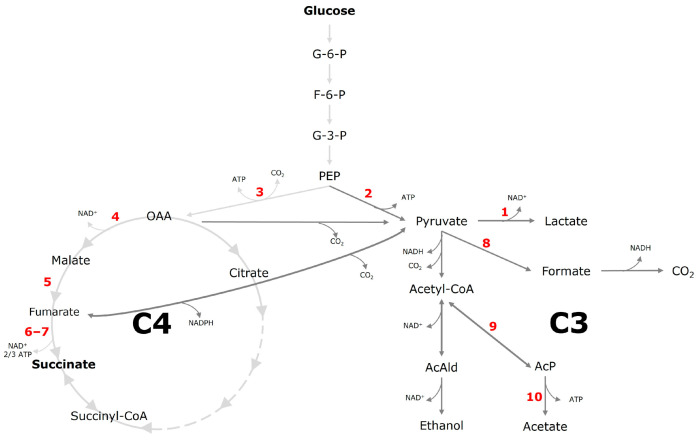
Putative metabolic pathway of *A. succinogenes* starting from glucose to the target product succinate and other by-products. The C4 metabolism is defined from oxaloacetate. The C3 metabolism is defined from pyruvate. Red numbers indicate enzymatic steps of the enzymes listed in Table 1. Alternative metabolic pathways which are not related to the products discussed in this paper are not shown, for simplification. Abbreviations: G-6-P, glucose-6-phosphate; F-6-P, fructose-6-phosphate; G-3-P, glyceraldehyde-3-phosphate; PEP, phosphoenolpyruvate; AcAld, acetaldehyde; AcP, acetyl phosphate; OAA, oxaloacetate. Scheme imported and modified from Tix et al. (2024) [28].

**Table 1 microorganisms-13-01720-t001:** Overview of the NAD^+^/NADH balance in the metabolic pathway for the production of succinate in *A. succinogenes*.

Pathway	NADH Balance
Glycolysis (per glucose)	+2 NADH
Pentose phosphate pathway (oxidative part)	+1 NADPH
C4-Pathway (Succinate)	−2 NADH
C3-Pathway (Acetate)	+1 NADH
C3-Pathway (Formate)	±0 NADH
C3-Pathway (Ethanol)	−2 NADH
C3-Pathway (Lactate)	−1 NADH

**Table 2 microorganisms-13-01720-t002:** Comparison of the normalized abundances of key metabolic enzymes of C4 and C3 metabolism in fermentations with different applied electric potentials.

No.	Enzymes	Ratio Abundances (1-Chamber) −1500/−1000 (mV)	Ratio Abundances (2-Chamber) −1500/−1000 (mV)	Gene ID
**1**	D-lactate dehydrogenase	2.93	2.94	Asuc_0005
**2**	Pyruvate kinase	0.90	0.96	Asuc_1171
**3**	Phosphoenolpyruvate carboxykinase (ATP)	1.08	1.01	Asuc_0221
**4**	Malate dehydrogenase, NAD-dependent	1.13	0.79	Asuc_1612
**5**	Fumarate hydratase	1.02	0.92	Asuc_0956
**6**	Fumarate reductase (flavoprotein subunit)	1.08	1.02	Asuc_1813
**7**	Fumarate reductase iron–sulfur subunit	1.21	0.99	Asuc_1814
**8**	Formate acetyltransferase	0.67	0.83	Asuc_0207
**9**	Phosphate acetyltransferase	0.96	1.09	Asuc_1662
**10**	Acetate kinase	0.92	0.99	Asuc_1661

## Data Availability

The original contributions presented in this study are included in the article/Appendix A. Further inquiries can be directed to the corresponding author.

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
