# Peer review of "Actinobacillus succinogenes in Bioelectrochemical Systems: Influence of Electric Potentials and Carbon Fabric Electrodes on Fermentation Performance"

_microorganisms, 2025, doi:10.3390/microorganisms13081720_

Round 1

Reviewer 1 Report

Comments and Suggestions for Authors

The manuscript “Actinobacillus succinogenes in bioelectrochemical systems: Influence of electric potentials and carbon fabric electrodes on fermentation performance” presents a cogent overview of succinate as a high-value platform chemical, highlighting its diverse applications. The biological background on Actinobacillus succinogenes is well summarised: its incomplete TCA cycle, reliance on the reductive C4 branch for succinate, and the redox balancing role of NADH/NAD⁺. The discussion of electrobiotechnology and distinguishing microbial electrosynthesis from electro-fermentation is clear and establishes the conceptual framework.

In introduction section:

I suggest revising Section 1.3 to include only information pertinent to the article’s objective, excluding definitions drawn from the literature.

Emphasise novelty by contrasting prior work (e.g. Park & Zeikus 1999; Peng et al. 2024) with the present focus on water-electrolysis thresholds and carbon fabric variants.

In materials and methods section:

Provide details on statistical analyses: normality tests, ANOVA assumptions, number of centre-point replicates in RCCD, and software versions.

Clarify the criteria for selecting the five carbon fabrics and the rationale for the chosen working electrode area (17.5 cm²).

I suggest including phenotypic or genotypic parameters to justify its selection (e.g. genes associated with extracellular electron transfer, hydrogen-production capacity or a metabolic profile favouring EET).

In results section:

Measured resistivity spans 7.1 × 10⁻³ to 2.7 × 10⁻³ Ω·m, establishing ACC-5092-15 as a compromise between conductivity and biofilm support. The authors may wish to provide measurements of the electrical current generated in the bioelectrochemical system (for example, current density under chronoamperometric conditions). Nor are there any cyclic voltammetry profiles showing redox signals attributable to endogenous mediators or to direct electron transfer.

It is strongly recommended to include at least one cyclic voltammogram measurement to characterise the redox processes involved in the biofilm. An electrochemical impedance spectroscopy (EIS) assessment would also be ideal to evaluate the biofilm–electrode interface, taking into account the proportional effect of the applied potentials.

In the absence of these elements, it is not possible to ascertain that A. succinogenes is effectively electrogenic or to describe how it establishes communication with the electrode—an aspect essential for the consideration of a bioelectrochemical system.

Line 402, review units “54.81 ± 2.28 °C” ¿centigrade? ¿temperature or Coulombs”

In conclusion section:

Propose specific next steps—e.g. continuous-flow BES trials, techno-economic assessments, or genetic strategies to channel excess reducing equivalents—so as to guide future work.

In general, ensure uniform abbreviation of compounds (e.g. NADH/NAD⁺, H₂O₂).

Comments on the Quality of English Language

Prefer active constructions and consistent past tense for methods/results (“We measured”, “Hydrogen evolved”) and present tense for established facts.

Conciseness: “It had been assumed that…” vs. “It was assumed that…”, or “This investigation provides important findings…” vs. “This study demonstrates…”.

Author Response

In introduction section:

I suggest revising Section 1.3 to include only information pertinent to the article’s objective, excluding definitions drawn from the literature.

Response: We would like to thank Reviewer 1 for the constructive suggestion. To facilitate the understanding of the contents for readers without a specific background in electrobiotechnology, we consider an introductory chapter with basic definitions of the various applications to be extremely helpful. It provides valuable orientation and enables better understanding of the results presented. For this reason, we have decided to keep the relevant subchapter 1.3.

Emphasise novelty by contrasting prior work (e.g. Park & Zeikus 1999; Peng et al. 2024) with the present focus on water-electrolysis thresholds and carbon fabric variants.

Response: We sincerely thank the reviewer for this valuable comment. We fully agree that highlighting the novelty of our approach in contrast to previous studies strengthens the introduction. In response, we have revised the final paragraph of the introduction to explicitly contrast our study with the works of Park & Zeikus (1999) and Peng et al. (2024). Specifically, we now emphasize the dual focus of our work: first, the use of low-cost carbon fabric electrodes as alternative electron donors to reduce material costs; and second, the deliberate application of electric potentials near the water electrolysis threshold to promote in situ hydrogen evolution and its potential impact on the reductive metabolism of A. succinogenes.

We hope this revision improves the clarity and originality of the manuscript.

In materials and methods section:

Provide details on statistical analyses: normality tests, ANOVA assumptions, number of centre-point replicates in RCCD, and software versions.

Response: Thank you for your comment regarding the statistical analyses. In our study, we did not perform inferential statistical tests such as ANOVA or normality tests, as our analysis was limited to descriptive statistics. Specifically, we calculated standard deviations based on the indicated number of replicates (n), which are clearly reported in the figure legends and methods section. These values reflect the variability within our experimental measurements and were determined using standard error propagation methods. As such, we believe that a more detailed discussion of statistical test assumptions is not applicable in this context, and no changes to the manuscript text are required. However, we would be happy to clarify this point in the revised version if the reviewer considers it necessary. Additionally, we note that a false discovery rate (FDR) of ≤1% was applied for the proteomics data analysis, as described in the Supplementary Information.

Clarify the criteria for selecting the five carbon fabrics and the rationale for the chosen working electrode area (17.5 cm²).

Response: The reason for choosing carbon fabrics as electrode material is the high specific surface area combined with simple application, as already described in lines 180-190. The surface area of 17.5 cm2 in the bench-top experiments in 130 mL reactor volume has proven to be the best size in previous experiments. At this size, the WE and CE do not come into contact with each other in the reactor and still provide a large surface area. To clarify this, the text in lines 235-236 has been adapted.

I suggest including phenotypic or genotypic parameters to justify its selection (e.g. genes associated with extracellular electron transfer, hydrogen-production capacity or a metabolic profile favouring EET).

Response: We thank Reviewer 1 for the suggestion. However, it is not clear from the comment which selection is specifically meant. If the comment refers to the selection of the production strain A. succinogenes, subchapter 1.4 should sufficiently clarify why this strain is of particular interest for the described application. We do not consider further embedding of phenotypic or genotypic parameters to be necessary in this context, as it would exceed the scope of the manuscript.

In results section:

Measured resistivity spans 7.1 × 10⁻³ to 2.7 × 10⁻³ Ω·m, establishing ACC-5092-15 as a compromise between conductivity and biofilm support. The authors may wish to provide measurements of the electrical current generated in the bioelectrochemical system (for example, current density under chronoamperometric conditions). Nor are there any cyclic voltammetry profiles showing redox signals attributable to endogenous mediators or to direct electron transfer.It is strongly recommended to include at least one cyclic voltammogram measurement to characterise the redox processes involved in the biofilm. An electrochemical impedance spectroscopy (EIS) assessment would also be ideal to evaluate the biofilm–electrode interface, taking into account the proportional effect of the applied potentials. In the absence of these elements, it is not possible to ascertain that A. succinogenes is effectively electrogenic or to describe how it establishes communication with the electrode—an aspect essential for the consideration of a bioelectrochemical system.

Response: We acknowledge the interesting comment to include CV or EIS measurements to complement the results. However, the main aim of the manuscript was not to investigate the exact mechanism of electron transfer of A. succinogenes, but to analyze the influence of different electrode materials and electrical potentials on product formation. A further investigation of the electrochemical activity of the organism by means of CV measurements is therefore not the focus of this article. In addition, the electroactive property of A. succinogenes and its influence on succinate production has already been described and confirmed in detail in several studies (e.g. Park & Zeikus, 1999; Wang et al., 2017; Pateraki et al., 2023; Peng et al., 2024).

Line 402, review units “54.81 ± 2.28 °C” ¿centigrade? ¿temperature or Coulombs”

Response: We thank the reviewer for pointing out this ambiguity. The unit "C" in this context refers to Coulombs, not degrees Celsius. To avoid confusion, we have revised the manuscript to explicitly state "C (Coulombs)" in line 402.

In conclusion section:

Propose specific next steps—e.g. continuous-flow BES trials, techno-economic assessments, or genetic strategies to channel excess reducing equivalents—so as to guide future work.

Response: We thank the reviewer for this valuable suggestion. In response, we have expanded the conclusion to include concrete future research directions. These comprise the evaluation of carbon fabric electrodes in continuously operated long-term bioelectrochemical systems, as well as the use of targeted genetic engineering strategies in A. succinogenes to suppress competing C3 pathways and to more efficiently channel electrochemically supplied reducing equivalents toward succinic acid production.

In general, ensure uniform abbreviation of compounds (e.g. NADH/NAD⁺, H₂O₂).

Response: Completed as described. All NADH/NAD+ have been standardized to NAD+/NADH and formatting errors of superscript and subscript numbers have been corrected.

Reviewer 2 Report

Comments and Suggestions for Authors

This manuscript presents a comprehensive investigation into how varying electric potentials and carbon fabric electrodes affect succinate production by Actinobacillus succinogenes in bioelectrochemical systems (BES). The study integrates electrochemical, microbiological, and proteomic analyses to decipher shifts in metabolite production and redox balance. The topic is timely and relevant for the development of sustainable bioprocesses involving electro-fermentation. Major Strengths.•    Original Contribution: The work addresses a novel intersection of electrochemistry and microbial fermentation, exploring the intracellular effects of redox manipulation.
•    Multi-Methodological Approach: The combination of BES performance data, biofilm microscopy, electrochemical measurements, and proteome analysis offers a well-rounded and data-rich view.
•    Clarity in Objectives: The study clearly defines its goal to optimise succinate production by assessing how potentials and electrode types influence fermentation.
•    Useful Comparative Design: Inclusion of both single- and two-chamber BES setups enhances the robustness of the findings.

Questions for the Authors
1. How were the Faraday efficiencies calculated? Was gas leakage or electrode surface loss considered?
2. What specific controls were applied to isolate the effects of the electric potential from other variables?
3. Could the observed decrease in biomass at highly negative potentials be due to toxicity from electrochemical byproducts?
4. How was biofilm accumulation distinguished from medium-derived deposits on the electrode?
5. What was the rationale for selecting the specific potentials tested (e.g., -1333 mV, -1666 mV)?

Author Response

Questions for the Authors

1.How were the Faraday efficiencies calculated? Was gas leakage or electrode surface loss considered?

Response: We thank the reviewer for this important question. The Faraday efficiency (FE) was calculated by quantifying the amount of hydrogen produced (in mmol), converting this value into charge using Faraday’s constant and comparing it to the total charge passed into the system as recorded by the potentiostat. This approach yields the ratio of Coulombs used for hydrogen evolution to the total Coulombs applied, i.e., the Faraday efficiency. Regarding potential gas leakage, we would like to emphasize that the experiments were conducted in a gas-tight electrochemical setup that was specifically developed and validated within our research group for hydrogen quantification. Therefore, we are confident that gas leakage and electrode surface degradation did not affect the accuracy of the Faraday efficiency determination.

  1. What specific controls were applied to isolate the effects of the electric potential from other variables?

Response: Control experiments with exactly the same setup without applied potential were used for comparison (see e.g. Figure 4A).

  1. Could the observed decrease in biomass at highly negative potentials be due to toxicity from electrochemical byproducts?

Response: Yes, the assumption is plausible. However, since we do not have detailed analyses in this context and only HPLC results available, we can only confirm a general inhibition of metabolic activity (see lines 440–442).

  1. How was biofilm accumulation distinguished from medium-derived deposits on the electrode?

Response: Preliminary tests have shown that the adsorption of media components was a minor factor in the biofilm determination.

  1. What was the rationale for selecting the specific potentials tested (e.g., -1333 mV, -1666 mV)?

Response: The selection of the specific potentials is dependent on the maximum capacity utilization of the experimental setup. The allocation of potentials should cover all ranges between 0 and -2000 mV with similar intervals.

Reviewer 3 Report

Comments and Suggestions for Authors

Reviewer Recommendation and Comments for manuscript microorganisms-3696724 with the title: “Actinobacillus succinogenes in bioelectrochemical systems: Influence of electric potentials and carbon fabric electrodes on fermentation performance”, authors: J. Tix, J.-N. Hengsbach, J. Bode, F. Pedraza, J. Willer, S.J. Park, K.F. Reardon, R. Ulber, N. Tippkötter.

Overall, the research contains novel scientific findings obtained through the employment of contemporary investigative methods. The data garnered hold significance for specialists in the fields of bioelectrochemistry. The work merits publication in the Microorganisms journal upon addressing the comments and responding to the questions.

The main comments that I find useful for improving the quality of the article are presented below:

‴ According to the experimental part, ACC-5092-10, -15, -20 and -25 with different specific surface areas, 1200, 1500, 1750 and 2100 m2g-1, were used. According to figure 1, the data for ACC-5092-10, -15, -20 and -25 are presented. The biofilm analysis (see lines 325, 326) indicates results for ACC-5092-5, -11, -15 and -25!? Why? Are there studies on other materials? Contradictory statements lead to confusion in understanding the results.

‴ line 337/ “Activation can optimize the pore structure of the carbon material, resulting in an increased specific surface area.” All 4 types of carbon fabric were treated/activated in the same way. Therefore, all 4 types of carbon fabric should have the same surface activity. The difference between them lies in the active surface area. The increase in biomass is attributed to the increase in the active surface area.

‴ line 351/ “The electrical conductivity of the various carbon fabrics was measured (Figure 2A).” There are differences between electrical conductivity, electrical resistivity and resistance. The authors did not measure either conductivity (S/m) or resistivity (Ωm), but they did measure resistance (Ω). The authors must clearly and correctly present the experimentally measured parameters.

‴ line 353/ “20 mm x 15 mm, 20 mm x 30 mm, 20 mm x 45 mm,” Are these surfaces exterior surfaces or cross sections? On which sides were the resistances measured? (20, 30 and 45 mm?)

‴ Figure 2/ Do the electrical parameters correspond to carbon fabrics coated with biofilm or without biofilm? Are there comparative data of the electrical parameters for carbon fabrics coated with biofilm and those not coated with biofilm?

‴ As the potential difference increases to values ​​of 1.8 or 2.0 V, associated processes of discharge of other biomolecules present in the system must also be taken into account. Not only water electrolysis. Possible electrolysis of glucose? Possible electrolysis of succinate? Possible electrolysis of lactate? For example, what is the stability of the biofilm at these high potentials, 1.8 or 2.0 V? Was the biomass determination done after electrolysis? (similar to 3.1 Biofilm analysis but after electrolysis). As the potential difference increases, A. succinogenes is destroyed!?

‴ Figure 7 has already been published (https://doi.org/10.3390/fermentation10100504). With the consent of the authors, this figure should be deleted.

‴ The authors cannot make any reference to intracellular redox potential or cell potential. These are mere speculations, without any experimental scientific basis. The authors have not measured such a potential. Or have they measured it?

‴ The typos must be corrected.

NaHPO4

l or L (volume unit should be  L)

g·l-1 / mg·ml-1 / concentration unit should be the same

H2

7.1·10-3 superior exponent and other

etc.

‴ The Microorganisms journal require a specific format of references, authors must pay more attention in their writing.

‴  There are some grammar and typing mistakes.

‴ The authors must revise the entire manuscript.

Comments on the Quality of English Language

The English could be improved to more clearly express the research.

Author Response

The main comments that I find useful for improving the quality of the article are presented below:

‴ According to the experimental part, ACC-5092-10, -15, -20 and -25 with different specific surface areas, 1200, 1500, 1750 and 2100 m2g-1, were used. According to figure 1, the data for ACC-5092-10, -15, -20 and -25 are presented. The biofilm analysis (see lines 325, 326) indicates results for ACC-5092-5, -11, -15 and -25!? Why? Are there studies on other materials? Contradictory statements lead to confusion in understanding the results.

Response: We thank the reviewer for carefully identifying this inconsistency. The mention of ACC-5092-5 and -11 was a typographical error. As correctly stated in the experimental section and shown in Figure 1, only ACC-5092-10, -15, -20, and -25 were used in this study. We have corrected the affected section accordingly and apologize for the confusion this may have caused.

‴ line 337/ “Activation can optimize the pore structure of the carbon material, resulting in an increased specific surface area.” All 4 types of carbon fabric were treated/activated in the same way. Therefore, all 4 types of carbon fabric should have the same surface activity. The difference between them lies in the active surface area. The increase in biomass is attributed to the increase in the active surface area.

Response: We thank the reviewer for this valuable comment and fully agree that the wording needed clarification. All four carbon fabrics were indeed subjected to the same activation procedure by the manufacturer. The differences observed in biomass formation are therefore not due to variations in treatment, but rather to material-specific differences in the resulting active surface area. To reflect this more accurately and avoid confusion, we have revised the term "specific surface area" to "active surface area" in the manuscript where appropriate.

‴ line 351/ “The electrical conductivity of the various carbon fabrics was measured (Figure 2A).” There are differences between electrical conductivity, electrical resistivity and resistance. The authors did not measure either conductivity (S/m) or resistivity (Ωm), but they did measure resistance (Ω). The authors must clearly and correctly present the experimentally measured parameters.

Response: We thank the reviewer for pointing out this important distinction. We fully agree that it is essential to use the correct terminology when referring to electrical properties. In response, we have revised the manuscript to clearly state that we measured the electrical resistance (Ω) of the carbon fabric samples. Furthermore, we now describe how specific resistance (Ω·cm) was calculated based on the measured values and the sample geometry, including the thickness of each fabric type. We appreciate the reviewer’s comment, which helped us improve the accuracy and clarity of our methodological description.

‴ line 353/ “20 mm x 15 mm, 20 mm x 30 mm, 20 mm x 45 mm,” Are these surfaces exterior surfaces or cross sections? On which sides were the resistances measured? (20, 30 and 45 mm?)

Response: We thank the reviewer for this helpful clarification request. The dimensions refer to the exterior surface area of the rectangular carbon fabric samples (length × width). The electrical resistance was measured along the longer edge of each sample (i.e., across 15, 30, and 45 mm, respectively), to assess how path length influences surface resistance. To further improve clarity and reproducibility, we have also added the corresponding fabric thicknesses used for calculating the resistivity. These were 0.66 mm for ACC-5092-10, 0.60 mm for -15, 0.55 mm for -20, and 0.54 mm for -25.

‴ Figure 2/ Do the electrical parameters correspond to carbon fabrics coated with biofilm or without biofilm? Are there comparative data of the electrical parameters for carbon fabrics coated with biofilm and those not coated with biofilm?

Response: We thank the reviewer for this important question. The electrical measurements presented in Figure 2 were performed prior to cultivation, using electrodes without biofilm coating. At this stage, the aim was to characterize the intrinsic electrical properties of the system and to better understand the influence of the different carbon fabric types on system resistance and potential distribution under defined baseline conditions.

‴ As the potential difference increases to values of 1.8 or 2.0 V, associated processes of discharge of other biomolecules present in the system must also be taken into account. Not only water electrolysis. Possible electrolysis of glucose? Possible electrolysis of succinate? Possible electrolysis of lactate? For example, what is the stability of the biofilm at these high potentials, 1.8 or 2.0 V? Was the biomass determination done after electrolysis? (similar to 3.1 Biofilm analysis but after electrolysis). As the potential difference increases, A. succinogenes is destroyed!?

Response: The biomass was also determined after fermentation, i.e. after electrolysis, as shown in Figure 5. This shows the influence of the applied potential and thus, particularly at higher potentials, the effect of electrolysis on the biomass both at the working electrode (WE) and in the remaining reactor volume. As assumed by Reviewer 3 in the last sentence of the comment, a negative influence on biomass formation at the working electrode and in the reactor volume can be observed at higher potentials, as already described in lines 484 to 490.We cannot say with certainty whether A. succinogenes is actually destroyed by the high potentials, as suggested by the reviewer, or whether its activity is simply inhibited by electrochemical side reactions. However, the HPLC results showed a clear decrease in glucose consumption (see Fig. 4A).

‴ Figure 7 has already been published (https://doi.org/10.3390/fermentation10100504). With the consent of the authors, this figure should be deleted.

Response: Figure 7 has been modified by adding the numbers for the presentation of the proteome results. It is clear from the caption that the figure is imported and modified from Tix et al. (2024).

‴ The authors cannot make any reference to intracellular redox potential or cell potential. These are mere speculations, without any experimental scientific basis. The authors have not measured such a potential. Or have they measured it?

Response: We agree with Reviewer 3 that the intracellular redox potential was not measured directly in this study. However, it is known from the literature that both an applied electrical potential and hydrogen can influence the NAD⁺/NADH ratio - and thus indirectly also the intracellular redox potential. Therefore, our statement on the possible change in the intracellular redox potential is not an unproven assertion, but an assumption supported by literature. Even if no direct measurement was carried out, the interpretation is based on well-documented correlations in specialist literature. (10.1128/jb.181.8.2403-2410.1999; 10.1002/elsc.202400053)

‴ The typos must be corrected.

NaHPO4

Response: Updated to Na2HPO4

l or L (volume unit should be  L); g·l-1 / mg·ml-1 / concentration unit should be the same

Response:

H2

Response: Updated to H2

7.1·10-3 superior exponent and other

Response: Revised in the complete document.

‴ The Microorganisms journal require a specific format of references, authors must pay more attention in their writing.

Response: Revised in the complete document.

‴ There are some grammar and typing mistakes.

Response: Revised in the complete document.

Reviewer 4 Report

Comments and Suggestions for Authors

The manuscript investigates the effects of different electric potentials and carbon fabric electrode materials on the metabolic performance of Actinobacillus succinogenes in bioelectrochemical systems. The goal is to optimize succinate production through electro-fermentation by manipulating intracellular redox balance, electrode composition, and applied voltage.

Suggestion:

Statistical analysis is mentioned (in lines 322, 545), but lacks details on tests used.

Author Response

Statistical analysis is mentioned (in lines 322, 545), but lacks details on tests used.

Response: Thank you for your comment regarding the statistical analyses. In our study, we did not perform inferential statistical tests such as ANOVA or normality tests, as our analysis was limited to descriptive statistics. Specifically, we calculated standard deviations based on the indicated number of replicates (n), which are clearly reported in the figure legends and methods section. These values reflect the variability within our experimental measurements and were determined using standard error propagation methods. As such, we believe that a more detailed discussion of statistical test assumptions is not applicable in this context, and no changes to the manuscript text are required. However, we would be happy to clarify this point in the revised version if the reviewer considers it necessary. Additionally, we note that a false discovery rate (FDR) of ≤1% was applied for the proteomics data analysis, as described in the Supplementary Information.

Round 2

Reviewer 1 Report

Comments and Suggestions for Authors

I agree with the author’s responses and with the revised version of the manuscript.

Author Response

Comment: I agree with the author’s responses and with the revised version of the manuscript.

Response: We sincerely thank the reviewer for their positive feedback and for taking the time to evaluate our revisions and responses. We truly appreciate the constructive comments that helped us improve the clarity and quality of the manuscript.

Reviewer 3 Report

Comments and Suggestions for Authors

The authors have taken into account the reviewer's comments and provided a revised and improved manuscript. I congratulate the authors on their work and consider that the manuscript can be considered for publication.

Author Response

Comment: The authors have taken into account the reviewer's comments and provided a revised and improved manuscript. I congratulate the authors on their work and consider that the manuscript can be considered for publication.

Response: 

We sincerely thank the reviewer for the thoughtful feedback, encouraging words, and for acknowledging the revisions we made. We are grateful for the constructive suggestions, which have significantly contributed to improving the quality and clarity of the manuscript.